# Elderly Suffering from ST-Segment Elevation Myocardial Infarction—Results from a Database Analysis from Two Mediterranean Medical Centers

**DOI:** 10.3390/jcm10112435

**Published:** 2021-05-30

**Authors:** Leor Perl, Alfonso Franzé, Fabrizio D’Ascenzo, Noa Golomb, Amos Levi, Hana Vaknin-Assa, Gabriel Greenberg, Abid Assali, Gaetano M. De Ferrari, Ran Kornowski

**Affiliations:** 1Department of Cardiology, Rabin Medical Center—Beilinson Hospital, Petach Tikva 4941492, Israel; noagolomb@gmail.com (N.G.); amos.levi@gmail.com (A.L.); hana100niki@gmail.com (H.V.-A.); gabrigr75@gmail.com (G.G.); ran.kornowski@gmail.com (R.K.); 2Sackler Faculty of Medicine, Tel Aviv University, Tel Aviv 69978, Israel; assali@clalit.org.il; 3Division of Cardiology, Department of Medical Sciences, Città della Salute e della Scienza, University of Turin, Corso Bramante 88, 10126 Turin, Italy; franze.alfonso@gmail.com (A.F.); fabrizio.dascenzo@gmail.com (F.D.); 4Department of Cardiology, Meir Medical Center, Tchernichovsky St 59, Kfar-Saba 4428164, Israel; 5Department of Cardiology, Fondazione IRCCS Policlinico San Matteo, Viale Camillo Golgi 19, 27100 Pavia, Italy; gaetanomaria.deferrari@unito.it

**Keywords:** octogenarians, elderly, myocardial infarction

## Abstract

**Background**: Little is known regarding primary percutaneous coronary intervention (pPCI) for ST-segment elevation myocardial infarction (STEMI) in the elderly. **Methods:** Data on 319 octogenarians, 641 septuagenarians, and 2451 younger patients was collected from an ongoing prospective registry of patients treated with pPCI for STEMI at two Mediterranean-area medical centers in 2009–2017. **Results**: More octogenarian patients were female (40.8 vs. 31.9 septuagenarians and 26.5% under 70 y, *p* < 0.01), had hypertension (79.5 vs. 69.5 and 45.9%, *p* < 0.01), renal failure (32.5 vs. 20.1 and 5.2%, *p* < 0.01), and a lower left-ventricular ejection fraction (42.0 vs. 44.9 and 47.6%, *p* = 0.012). At 1 month and 3 years after intervention, mortality was higher in the octogenarian patients (12.2 vs. 7.9%, *p* = 0.01; and 36.7 vs. 23.1%, *p* < 0.01, respectively), with no significant differences in the rates of recurrent myocardial infarction, target vessel revascularization, coronary artery bypass surgery, and cardiovascular death. Following adjustment for confounders, 3-year mortality was significantly higher in the octogenarians (HR 3.89 vs. 3.19 for septuagenarians, *p* < 0.01), but rates of major adverse cardiac events or cardiovascular death were not. **Conclusions**: Despite suffering from higher all-cause mortality, octogenarian patients treated with pPCI for STEMI do not suffer an increased risk of ischemic cardiac events relative to younger patients.

## 1. Introduction

The general population of the developed world is gradually ageing, and the proportion of octogenarians is expected to triple by 2050 [1]. Cardiovascular disease poses a growing burden as age increases, and it remains the leading cause of morbidity and mortality in the elderly. Indeed, age is the strongest risk factor for the development of coronary heart disease [1,2,3]. In the United States, approximately one-third of deaths in the elderly each year can be attributed to acute coronary syndromes [2].

Compared to younger patients, the elderly present with more comorbidities and are at higher risk of mortality and complications following percutaneous coronary intervention (PCI) [4], especially those with acute coronary syndromes or ST-elevation myocardial infarction (STEMI) [5,6,7,8,9,10,11,12,13]. Furthermore, elderly patients are more likely to present with atypical symptoms, increasing the risk of delay in treatment or misdiagnosis [14,15]. Nevertheless, the benefit of revascularization in the elderly population is well established [9,16,17,18,19], and PCI is associated with better outcomes and a lower rate of bleeding than fibrinolysis [20,21,22]. Both the European and American guidelines emphasize that there is no upper age limit for reperfusion, and in the elderly population, as in younger patients, an early invasive strategy is preferred [23,24].

The proportional representation of elderly patients in clinical trials assessing revascularization for STEMI is much lower than in routine clinical practice [25,26,27]. Therefore, information on the clinical characteristics and outcomes of octogenarians undergoing revascularization, particularly by means of primary PCI (pPCI), is limited. The aim of the present study was to examine the short- and long-term outcomes of octogenarian patients who were treated with pPCI for STEMI at two large Mediterranean-area tertiary medical centers.

## 2. Materials and Methods

### 2.1. Patients and Setting

The clinical data of consecutive patients presenting with STEMI and treated by pPCI at the Department of Cardiology, Rabin Medical Center, Israel, between January 2009 and December 2017 or the Department of Medical Sciences, University of Turin, Italy, between June 2015 and January 2017 were prospectively entered into a registry for purposes of monitoring patient-related parameters and clinical events. Patients included in this post hoc study were divided into octogenarian, septuagenarian, and younger than 70 years of age. Exclusion criteria were presentation with cardiogenic shock, treatment by thrombolysis, and ineligibility for a year-long dual antiplatelet regimen. The study was approved by the Institutional Review Boards of the two participating centers (study number RMC-3741 at the Rabin Medical Center and study number 0119191 at the Department of Medical Sciences, University of Turin).

### 2.2. Interventional Procedure

All patients provided explicit written informed consent before undergoing cardiac catheterization. Pretreatment consisted of aspirin and unfractionated heparin (70 U/kg); clopidogrel 300 or 600 mg, prasugrel 60 mg, or ticagrelor 180 mg was administered as a loading dose before or immediately after PCI. The utilization of glycoprotein IIb/IIIa inhibitors and the choice of drug-eluting stent versus bare-metal stent were left to the discretion of the primary operator. All stents were implanted with moderate-to-high deployment pressure (14 to 18 atm). All patients received dual antiplatelet therapy with aspirin 100 mg daily and a P2Y12 inhibitor (clopidogrel, prasugrel, or ticagrelor) for at least 12 months after PCI.

### 2.3. Study Endpoints

Immediate and in-hospital clinical events were prospectively recorded in the institutional database. During follow-up, patients completed standardized questionnaires on clinical events at 6-month intervals either by telephone or in the outpatient clinic. When indicated, records from peripheral hospitals were acquired to verify the events. All events were further confirmed and adjudicated by the institutional clinical events adjudication committees comprising of three independent physicians. Survival status was assessed by municipal civil registries at 1 and 3 years.

Clinical outcomes included all-cause mortality and major adverse cardiac events (MACE), which comprised myocardial infarction (MI), target vessel revascularization (TVR), and coronary artery bypass surgery (CABG). Other endpoints were peri-procedural arrhythmias, vascular complications (periprocedural hematomas, retroperitoneal bleeding, pseudoaneurysms, arteriovenous fistula, arterial thrombosis, distal embolism, dissection, and transient limb ischemia), 30-day bleeding according to the thrombolysis in myocardial infarction (TIMI) classification, acute kidney injury (defined as a serum creatinine elevation ≥50% or ≥0.3 mg/dL from baseline in the first 72 h), stent thrombosis, and cardiac death (defined as death occurring due to cardiovascular causes or cerebrovascular causes and any death without another known cause). Anemia was defined as hemoglobin levels lower than 13.0 g/dL for men and 12.0 g/dL for women. Renal failure at baseline was defined as glomerular filtration rate below 50 mL/min/1.73 m^2^ (calculated according to the Modification of Diet in Renal Disease formula). Findings were compared between patients younger than 70 years, septuagenarian, and octogenarian patients.

### 2.4. Statistical Analysis

Continuous data are summarized as mean and standard deviation (SD) or median and interquartile range (IQR), and categorical data as frequency (%). Analysis of variance (ANOVA) was used to compare continuous variables between groups, and chi-square or Fisher’s exact test was used for categorical variables. The normality of variable distributions was assessed using the Kolmogorov–Smirnov test. Time-to-event curves were constructed using the Kaplan–Meier method and compared using a log-rank test. In addition, competing risk analysis was performed, presented as cumulative incidence function. Cox regression analyses were performed to identify independent predictors of the primary end point. Effect sizes are presented as odds ratio and 95% confidence interval (CI). Stepwise variable selection of significant univariate predictors (*p* < 0.1) was used to identify variables for inclusion in the multivariate model. Multivariate logistic regression analysis was performed to determine independent predictors of the primary end point, accounting for known baseline cardiovascular risk differences. All statistical analyses were performed with IBM SPSS V.27 except for the competing risk analysis, which was performed with R V.4.0.0 software. A *p* value of <0.05 was considered statistically significant.

## 3. Results

A total of 3411 patients were included in the STEMI registry at the time of the study: 2828 from Rabin Medical Center and 583 from the University of Turin Medical Center. Of these, 960 consecutive patients (28.1%) were older than 70 years. There were 641 septuagenarians of mean age 75.3 ± 2.8 years, 319 octogenarians of mean age 85.5 ± 3.6 years, and 2451 patients younger than the age of 70 (mean age 55.8 ± 8.7). Baseline characteristics, procedural details, and quantitative coronary angiographic data were available for all of them.

The clinical and treatment-related characteristics of the different age groups are shown in Table 1 and Table 2. The octogenarian group had a higher proportion of female patients, when compared with the septuagenarians and younger patients (40.8 vs. 31.9 and 26.5%, respectively, *p* < 0.01) and higher rates of diabetes (35.6 vs. 37.2 and 24.5%, *p* = 0.04), hypertension (79.5 vs. 69.5 and 45.9%, *p* < 0.01), previous CABG (7.1 vs. 4.1 and 2.5%, *p* < 0.01), renal failure (32.5 vs. 20.1 and 5.2%, *p* < 0.01), previous CVA (12.9 vs. 13.6 and 3.6%, *p* = 0.023), and peripheral artery disease (10.0 vs. 7.1 and 4.0%, *p* < 0.01). However, mean body mass index was lower (26.7 ± 5.2 vs. 27.2 ± 5.6 and 28.1 ± 6.0, *p* = 0.042). Older patients presented with a lower left-ventricular ejection fraction (42.0 vs. 44.9 and 47.6%, *p* = 0.012), and more had severely calcified (32.6 vs. 25.3 and 27.5%, *p* = 0.021) and bifurcation lesions (35.8 vs. 28.3 and 27.3%, *p* = 0.011). They were more often treated with clopidogrel (55.8 vs. 48.7 and 14.8%, *p* = 0.039) or intra-aortic balloon pump (5.5 vs. 6.1 and 2.9%, *p* = 0.001), but not with drug-eluting stents (82.1 vs. 84.3 vs. 92.1%, *p* = 0.014). There were no significant differences between the groups in rates of periprocedural complications, including vascular events, acute renal failure, and arrhythmias (Table 1 and Table 2).

Outcome analyses showed that compared to the septuagenarian group and patients younger than 70 years of age, the octogenarians had a higher rate of all-cause death at 1 month (19.0 vs. 12.3 and 2.9%, *p* = 0.01), 12 months (27.4 vs. 19.3 and 4.7%, *p* < 0.01), and 3 years after treatment (38.9 vs. 25.7 and 8.2%, *p* < 0.01) (Figure 1). At 3 years, there was no between-group difference in the rate of cardiovascular death (9.6 vs. 8.8 and 6.9%, *p* = 0.12); rates of recurrent MI, TVR, CABG, and MACE (Figure 1 and Figure 2); or rate of stent thrombosis at 3 years (2.1 vs. 3.6 and 2.9%, *p* = 0.13). A competing risk analysis for the causes of death also showed differences in noncardiovascular and all-cause death, but not cardiovascular death (Figure 3).

Univariate analysis identified the following predictors of 30-day and 3-year mortality after STEMI: age over 80, previous PCI, previous MI, previous CABG surgery, diabetes mellitus, hypertension, hyperlipidemia, renal failure, LVEF, the implantation of DES, and intra-aortic balloon pump (Table 3a,b). We used these factors, including age, gender, and the administration of Prasugrel as opposed to clopidogrel for the multivariate analysis. On regression analysis (Table 4a,b), factors significantly related to an increased risk of death at 1 month were age over 80 years (HR 1.88; 95% CI 1.02–4.78, *p* = 0.041), diabetes mellitus (HR 1.97; 1.04–3.72, *p* = 0.038), renal failure (HR 2.02; 1.08–3.80, *p* = 0.029), reduced left-ventricular ejection fraction (HR 0.93; 0.90–0.95 for each additional 1% in ejection fraction, *p* < 0.001), and the use of intra-aortic balloon pump (HR 1.79; 1.02–4.21, *p* = 0.032, Table 3a). At 3 years, the parameters significantly related to all-cause death were age group (HR 3.89; 2.74–5.52 for octogenarians, and 3.19; 2.33–4.36 for septuagenarians, *p* < 0.001 for both), diabetes mellitus (HR 1.41; 1.06–1.85, *p* = 0.016), renal failure (HR 2.44; 1.82–3.28, *p* < 0.001), left-ventricular ejection fraction (HR 0.94; 0.92–0.97, *p* < 0.001), and the implantation of drug-eluting stents (HR 0.66; 0.48–0.89, *p* = 0.008, Table 3b). The 30-day MACE was also assessed in regression analysis, demonstrating an increase in risk for events for patients with previous PCI (HR 1.42; 1.01–2.67, *p* = 0.042), diabetes mellitus (HR 2.45; 1.43–4.37, *p* = 0.021), and renal failure at baseline (HR 2.21; 1.23–4.98, *p* = 0.008). Age was not an independent predictor of 30-day MACE. A test for interaction of the individual medical center and outcomes yielded no significant results.

## 4. Discussion

Our study encompassed the experience of two Mediterranean-area tertiary medical centers with pPCI for the treatment of STEMI in the elderly. Compared to septuagenarians, octogenarians had a higher rate of all-cause mortality but similar rates of cardiac death, cardiac ischemic endpoints, and MACE.

There is relatively little information regarding the outcomes of octogenarians undergoing pPCI. Most of the evidence is derived from single-center observational studies. An early single-center study from the Netherlands that examined the outcomes of 4506 patients with STEMI reported a very high 1-year mortality rate (28.2%) in the octogenarian group (*n* = 379) [7]. Another, from Canada [5], compared 164 octogenarians with patients 65–69 years old and found that the octogenarian group was more likely to have a delay in treatment, increased rates of bleeding, acute kidney injury, and rehospitalization, and a trend toward longer hospital stay following pPCI. Their overall survival after 12 months was significantly lower than the control value (79% vs. 94%, *p* < 0.01). An additional study based on the Korean Acute Myocardial Infarction Registry (KAMIR) reported a significantly higher 12-month all-cause mortality rate in octogenarians than in nonoctogenarians (22.3% vs. 6.5%, *p* < 0.001) [11]. Additionally, a separate report based on the KAMIR registry demonstrated an incremental relationship with MACE. [28] However, two observational studies from China [20,29] showed that in elderly patients with STEMI, early reperfusion, especially with pPCI, was safe and effective compared with no reperfusion. Regarding the type of reperfusion, one of these studies assessed the use of fibrinolytic therapy in elderly patients with STEMI, with or without PCI, and found that it was associated with an eightfold increase in hemorrhagic stroke with no mortality benefit [22].

Our study is in good agreement with previous reports in terms of elevated all-cause death rates for the octogenarian group, reaching about 20% after 12 months. However, our experience shows that in survivors, the rates of other outcomes related to ischemic cardiac events, such as TVR, recurrent MI, and cardiac death, are similar between octogenarians and controls a decade younger. These findings partially contrast with the study from Korea, which suggested that age was incrementally related to MACE and cardiac death, although not to TVR [28]. Yet, they are in line with the separate analysis from the same registry, which found that octogenarian patients with acute MI had higher rates of mortality than nonoctogenarians, with no difference in rates of target lesion revascularization, TVR, recurrent acute MI, and CABG [11].

A higher rate of all-cause mortality is expected in the octogenarian and septuagenarian populations. We therefore assessed cardiovascular ischemic outcomes as well, such as cardiovascular death, TVR, recurrent MI, and CABG. It is important to mention that some of these outcomes may be influenced by the cardiac team’s preference of therapy (for example, rates of TVR or CABG may be somewhat lower in the very elderly patient population due to the higher perceived risk of the procedure). Nevertheless, the observation that other “hard” end points, such as cardiac death and MI, occurred at similar rates to controls, suggests similar benefits for pPCI for STEMI in the elderly. Hence, this study reinforces the present guidelines for the assessment and treatment of older patients with STEMI; namely, use of a similar approach to that applied in the general population in terms of timely revascularization preferably by pPCI when feasible [23,24]. At the same time, our study highlights the need for global evaluation of elderly patients with STEMI in order to correctly understand their prognosis, especially in cases in which survival appears to be influenced by noncardiologic conditions such as malignancies and neurological and degenerative diseases [13].

Regarding adjunctive therapy, some studies support a short duration of dual antiplatelet therapy for elderly patients treated by PCI due to a higher risk of bleeding [30,31]. All patients in our cohort were advised to continue with dual antiplatelet therapy for at least 12 months. This may have reduced the rate of ischemic outcomes in our study, while surprisingly not significantly raising the rate of bleeding in the octogenarians relative to the septuagenarians. The longstanding debate regarding the length of dual antiplatelet therapy in this patient population persists, although at least in patients with STEMI, our study adds evidence for the benefit of a 12-month-long treatment regimen in preventing recurrent ischemic events.

We opted to report on patients in the eighth and ninth decade of their lives. However, as the population ages, more nonagenarians and even centenarians may be considered for revascularization. There is already some evidence to support invasive therapy in these age groups, although the mortality rate is higher, even when compared with octogenarians [19,32]. In the future, studies assessing outcomes, as well as the preferred mode of both invasive and medical therapy for these patients, will shed more light on this growing patient population.

## 5. Limitations

The observational design of this study, as well as the arbitrary choice of the age of the control group, precludes speculations regarding ways to improve the prognosis of elderly patients treated with pPCI for STEMI. As such, we cannot rule out significant selection bias of both the study and control groups. A randomized controlled study comparing pPCI with conservative therapy in two similar elderly cohorts would more appropriately assess the true impact of revascularization in this patient population. However, at the present time, such a study design is likely to be considered unethical. In addition, we had information on whether the cause of death was cardiac or noncardiac, but specific details were missing. Therefore, the exact causes of the relatively high case-fatality rates in the octogenarians are not well defined. Despite these limitations, this study is one of the largest on elderly patients with STEMI treated by pPCI, derived from two tertiary medical centers, and it sheds important light on the prognosis of this growing patient population.

## 6. Conclusions

Patients more than 70 years old with STEMI who are treated by pPCI suffer from higher rates of all-cause death but similar rates of ischemic cardiac events, including cardiovascular death, TVR, recurrent MI, and CABG. These data support the importance of pPCI in cases of STEMI in the elderly population. Although future prospective trials assessing the efficacy of pPCI in the elderly population suffering from STEMI may presently be considered unethical, studies examining aspects of specific device and medical treatment modalities for this unique group of patients are warranted.

## Figures and Tables

**Figure 1 jcm-10-02435-f001:**
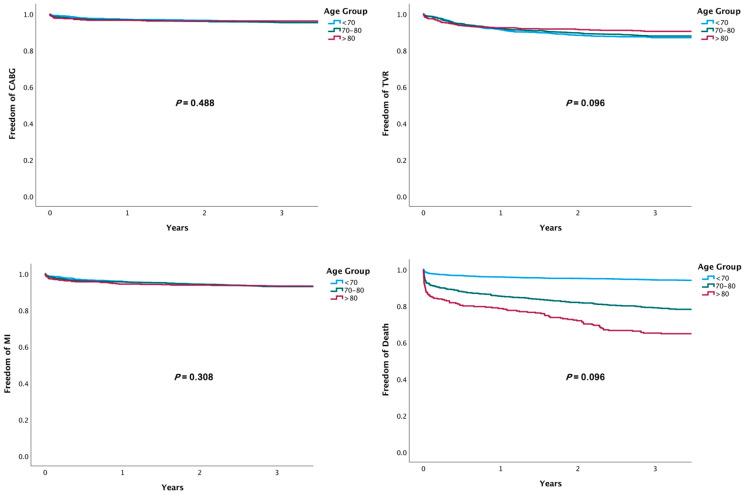
Kaplan–Meier curves of ischemic outcomes according to age.

**Figure 2 jcm-10-02435-f002:**
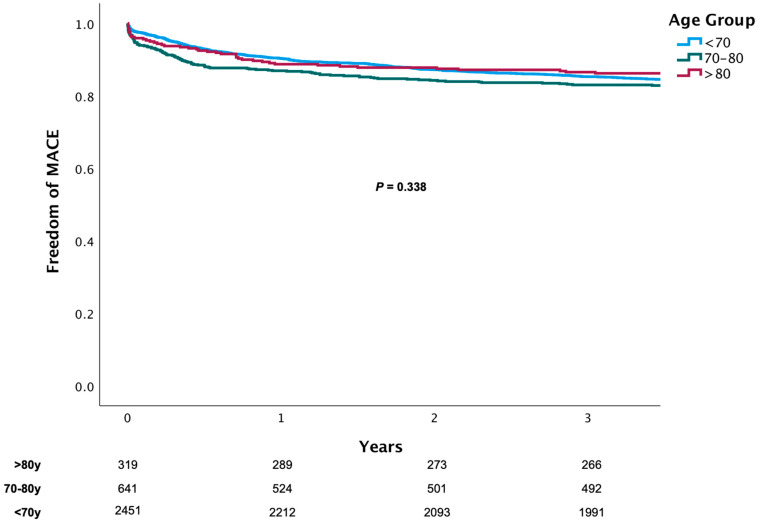
Kaplan–Meier curve of MACE according to age.

**Figure 3 jcm-10-02435-f003:**
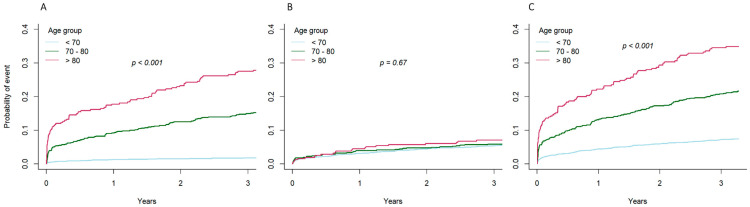
Cumulative incidence function of death according to age. (**A**) Probability of noncardiovascular death; (**B**) probability of cardiovascular death; (**C**) probability of all-cause death.

**Table 1 jcm-10-02435-t001:** Baseline characteristics.

Characteristic	80y and above (*n* = 319)	70–80y (*n* = 641)	70y and below (*n* = 2451)	*p* Value
Mean age (years)	85.5 ± 3.6	75.3 ± 2.8	55.8 ± 8.7	<0.001
Male (%)	59.2	68.1	73.5	0.040
BMI (kg/m^2^)	26.7 ± 5.2	27.2 ± 5.6	28.1 ± 6.0	0.042
T2DM (%)	35.6	37.2	24.5	0.040
Hypertension (%)	79.5	69.5	45.9	<0.001
Hyperlipidemia (%)	53.2	52.3	54.9	0.478
Smoking (%)				0.125
Active	32.2	33.5	31.2	
Past	11.2	9.8	8.1	
Never	56.6	56.7	60.7	
Anemia (%)	24.2	21.3	18.4	0.089
Previous PCI (%)	20.6	16.2	16.0	0.115
Previous myocardial infarction (%)	17.0	17.1	10.2	0.832
Previous CABG (%)	7.1	4.1	2.5	<0.001
Renal failure (%)	32.5	20.1	5.2	<0.001
Previous CVA (%)	12.9	13.6	3.6	0.023
Peripheral artery disease (%)	10.0	7.1	4.0	<0.001
Aspirin (%)	84.2	77.1	56.7	<0.001
Statins (%)	73.2	70.2	49.5	<0.001
Beta-blockers (%)	68.2	64.1	59.7	0.013
ACE inhibitors (%)	52.1	50.3	47.2	0.136
LVEF (%)	42.0 ± 12.1	44.9 ± 11.9	47.6 ± 11.7	0.012
GFR (mL/min/1.73 m^2^)	81.3 ± 19.1	84.6 ± 20.9	86.6 ± 22.3	0.002
Hemoglobin (g/dL ± SD)	12.1 ± 2.1	13.1 ± 2.2	13.5 ± 2.6	0.014
Platelets (× 10^9^/L. ± SD)	182.2 ± 45.1	183.3 ± 46.4	183.5 ± 50.1	0.138
Glucose (mg/L ± SD)	137.2 ± 52.7	133.1 ± 49.6	134.1 ± 52.1	0.241
Peak-CPK (mcg/L ± SD)	2085.7 ± 331.3	1986.5 ± 323.0	2123.7 ± 325.8	0.423

BMI—body mass index, T2DM—diabetes mellitus, PCI—percutaneous coronary intervention, CABG—coronary artery bypass graft surgery, CVA—cerebrovascular accident, ACE – angiotensin converting enzyme, LVEF—left ventricular ejection fraction, GFR—glomerular filtration rate, CPK - creatine phosphokinase.

**Table 2 jcm-10-02435-t002:** Procedural and in-hospital characteristics.

Characteristic	80y+ (*n* = 319)	70–80y (*n* = 641)	70y- (*n* = 2451)	*p* Value
Median door-to-balloon time (IQR)	1.09 (0.2–4.5)	1.04 (0.3–4.2)	1.03 (0.2–4.1)	0.301
Calcification (%)	32.6	25.3	27.5	0.021
Bifurcation lesions (%)	35.8	28.3	27.2	0.011
Mean number of affected vessels	1.9 ± 0.4	2.1 ± 0.5	1.8 ± 0.3	0.828
Stenting (%)	94.0	95.5	96.4	0.083
Culprit vessel (%)				0.542
Left main artery	4.5	4.6	5.1	
Left anterior descending artery	42.3	40.1	40.6	
Circumflex artery	20.2	22.1	24.7	
Right coronary artery	30.4	29.4	27.8	
Venous graft	2.6	3.8	1.8	
DES (%)	82.1	84.3	92.1	0.014
Stent diameter (mm)	3.0 ± 0.5	3.1 ± 1.0	3.1 ± 0.9	0.855
Stent length (mm)	20.2 ± 6.8	20.0 ± 7.4	19.5 ± 6.4	0.247
IABP (%)	5.5	6.1	2.9	0.001
Cardiac arrest (%)	0.7	0.8	0.8	0.728
Arrhythmias (%)	8.0	7.1	6.8	0.771
Vascular complications (%)	1.2	0.9	0.6	0.480
30-day TIMI major bleeding (%)	0.9	0.9	0.8	0.829
30-day TIMI minor bleeding (%)	2.1	2.0	1.8	0.121
Acute kidney injury (%)	12.9	9.1	8.8	0.093
Median admission length (IQR)	4.1 (1.8–6.9)	4.1 (1.6–7.2)	3.9 (1.1–5.4)	0.127
Aspirin (%)	90.7	90.3	92.8	0.107
Ticagrelor (%)	9.9	5.5	4.2	0.132
Clopidogrel (%)	55.8	48.7	14.8	0.039
Prasugrel (%)	3.8	42.1	79.5	<0.001
Statins (%)	94.5	96.5	97.2	0.192
Beta-blockers (%)	72.2	68.2	67.3	0.113
ACE inhibitors (%)	66.3	65.8	62.1	0.005

IQR—interquartile range, IABP—intra-aortic balloon pump, DES—drug eluting stents, TIMI—thrombolysis in myocardial infarction.

**Table 3 jcm-10-02435-t003:** (**a**) Univariate analysis of 30-day mortality predictors. (**b**) Cox regression analysis of 3-year mortality predictors.

Parameters	Hazard Ratio	Lower	Upper	*p*-Value
(**a**)				
Age group 70–80	1.38	0.93	2.34	0.261
Age group >80	1.57	1.07	4.11	0.002
Gender (female)	1.24	0.82	1.45	0.338
Previous PCI *	1.32	1.20	4.51	0.002
Previous myocardial infarction	1.09	1.00	1.92	0.080
CABG **	1.66	1.34	4.92	0.040
T2DM ^||^	1.49	1.02	3.98	0.021
Hypertension	1.09	1.01	2.02	0.092
Hyperlipidemia	1.20	1.03	2.98	0.004
Renal failure	2.73	1.45	4.98	0.007
LVEF ^¶^ (additional 1%)	0.91	0.67	0.98	<0.001
DES ^#^	0.67	0.38	1.95	0.082
Prasugrel	0.76	0.46	1.19	0.121
IABP ***	1.82	1.02	3.75	0.040
(**b**)				
Age group 70–80	1.97	0.89	5.31	0.110
Age group >80	3.62	1.83	6.23	<.001
Gender (female)	1.35	0.90	1.83	0.119
Previous PCI *	1.32	0.99	1.81	0.094
Previous myocardial infarction	1.42	1.02	2.32	0.083
CABG **	1.58	1.11	3.94	0.067
T2DM ^||^	1.62	1.06	4.29	0.006
Hypertension	1.08	0.97	4.01	0.072
Hyperlipidemia	1.23	1.00	3.29	0.100
Renal failure	2.67	1.63	5.29	<0.001
LVEF ^¶^ (additional 1%)	0.91	0.82	0.97	<0.001
DES ^#^	0.62	0.41	0.93	0.006
Prasugrel	0.72	0.41	1.49	0.231
IABP ***	1.61	1.00	4.12	0.090

* PCI—percutaneous coronary intervention, ** CABG—coronary artery bypass graft surgery, ^||^ T2DM—type 2 diabetes mellitus, ^¶^ LVEF—left ventricular ejection fraction, ^#^ DES—drug-eluting stent, *** IABP—intra-aortic balloon pump.

**Table 4 jcm-10-02435-t004:** (**a**) Cox regression analysis of 30-day mortality predictors. (**b**) Cox regression analysis of 3-year mortality predictors.

Parameters	Hazard Ratio	Lower	Upper	*p*-Value
(**a**)				
Age group 70–80	1.52	0.84	2.73	0.371
Age group >80	1.88	1.02	4.78	0.041
Gender (female)	1.29	0.96	1.81	0.255
Previous PCI *	0.93	0.39	2.20	0.871
Previous myocardial infarction	0.62	0.24	1.64	0.338
CABG **	1.66	0.56	4.97	0.364
T2DM ^||^	1.97	1.04	3.72	0.038
Hypertension	0.97	0.48	1.94	0.926
Hyperlipidemia	1.10	0.57	2.13	0.778
Renal failure	2.02	1.08	3.80	0.029
LVEF ^¶^ (additional 1%)	0.93	0.89	0.95	<0.001
DES ^#^	0.55	0.26	1.17	0.119
Prasugrel	0.78	0.42	1.82	0.729
IABP ***	1.79	1.02	4.21	0.032
(**b**)				
Age group 70–80	3.19	2.33	4.36	<0.001
Age group >80	3.89	2.74	5.52	<0.001
Gender (female)	1.24	0.94	1.65	0.127
Previous PCI *	1.13	0.77	1.67	0.524
Previous myocardial infarction	1.32	0.41	1.98	0.156
CABG **	1.53	0.91	2.56	0.111
T2DM ^||^	1.41	1.06	1.85	0.016
Hypertension	1.03	0.76	1.39	0.851
Hyperlipidemia	1.18	0.89	1.55	0.245
Renal failure	2.44	1.82	3.28	<0.001
LVEF ^¶^ (additional 1%)	0.94	0.92	0.97	<0.001
DES ^#^	0.66	0.48	0.89	0.008
Prasugrel	0.75	0.49	1.27	0.371
IABP ***	1.72	0.82	9.31	0.821

* PCI—percutaneous coronary intervention, ** CABG—coronary artery bypass graft surgery, ^||^ T2DM—type 2 diabetes mellitus, ^¶^ LVEF—left ventricular ejection fraction, ^#^ DES—drug-eluting stent, *** IABP—intra-aortic balloon pump.

## Data Availability

Anonymized study data can be requested by contacting Leor Perl (leorperl@gmail.com).

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
