# Peer review of "Elderly Suffering from ST-Segment Elevation Myocardial Infarction—Results from a Database Analysis from Two Mediterranean Medical Centers"

_jcm, 2021, doi:10.3390/jcm10112435_

Round 1
Reviewer 1 Report
Thank you for the opportunity to review this manuscript. The manuscript addresses an important topic. However, I recommend to take the following suggestions into consideration.
General points:
- In order to conclude that octogenarians do not suffer increased risk of ischemic cardiac events after PCI (line 39) a competing risk analysis should be added since it would be possible that some octogenarians already died before another PCI would be necessary. As it is shown they have a higher all-cause mortality.
- In addition, I would recommend not including “death” (line 100) in the MACE as it has already been analyzed as the primary endpoint.
- Please also explain in more detail if “other endpoints” such as vascular complications (line 102) were also included in MACE. I would suggest to include them and also include these endpoints in table 2 (e.g. bleeding is missing).
- “renal failure” in Line 103: Is it postprocedural renal failure? Please explain in more detail and also explain how renal failure was defined in patients with a chronic kidney disease and an eGFR<50ml/min at admission.
- “vascular complications”: is it perforation of a coronary artery? Please describe this in more detail (line 102).
- How was “cardiac death” (line 105) defined?
- Please describe inclusion criteria and endpoints in more detail: e.g. what is the institutional clinical events adjudication committee? Is the adjudication performed by physicians? (line 97).
- I would recommend to add a cox regression analysis for MACE at 30 days and 1-month.
- Baseline table: I would recommend to add laboratory parameters such as hemoglobin, electrolytes, hs-cTnT, NTproBNP, creatinine, BUN etc. at admission. In addition, I would also add data on medication at admission and discharge and data of the “culprit lesion” and if the patient suffers a more vessel disease. Additionally, I would also include some of these variables in the cox regression analysis.
- How was obesity in the baseline table defined? would suggest to replace it by BMI.
- It would be interesting to see data on the lengths of hospitalization of the different age groups. In addition, are there data on possible hospital acquired infections and frailty of patients?
Statistics:
- Several times is written p<0.01, but several p-values also have 3 decimal places. I would strongly recommend to show p-values in a uniform manner with 3 decimal places.
- Please explain the rational of p<0.1 for selection of univariate predictors (line 117) instead of using p<0.01.
- Please also show the univariable analyses of your Cox regression.
- Please review normal distribution of continuous variables in the baseline table: e.g. in door to ballon times the standard deviation is nearly the mean. Therefore some variables might not be normally distributed and a presentation as median would be preferable.
Minor points:
- Please state in the introduction to which of the both remaining group octogenarians are compared to (especially the order): Line 28-31.
- Please correct bifurcation to bifurcation lesions in table 2.
Reviewer 2 Report
In the manuscript by Perl et al., the authors investigated in a prospective registry study of two Mediterranean-area medical centers the importance of age on survival and further cardiovascular events in elderly patients suffering from myocardial infarction and treated with percutaneous coronary intervention. Cumulative 3411 patients were investigated in the study and divided in an octogenarian (+80 years), septuagenarian (70-80 years) and a younger group (<70 years). Interestingly the authors found out that the octogenarian patients, although suffering from a higher all-cause mortality, they did not have an increased risk for further ischemic cardiovascular events than the younger patients.
Due to the developments in the demographic change, this is a very interesting study investigating the importance of percutaneous coronary intervention in elderly patients with myocardial infarction. I have only a few concerns with the study, which I have outlined below:
- The authors reported that patients of two prospective registry studies (from Israel and Italy) were included in the analyzation. Ethical approval and study numbers should be added to the methods part.
- Table 1 demonstrates the baseline characteristics of the patients cohorts. In addition to the listed cardiovascular risk factors, smoking should be added to the investigation.
- In Table 2 procedural and in-hospital characteristics are listed. As an additional information, number of vessels affected by coronary artery disease should be added to the analyzation.
- I really appreciate the discussion part, in which the authors critically categorized their results in the current state of literature and medical knowledge. They state that there were no more bleeding events in the elderly patient population. Are there any detailed data on this from the study?
